

# Convolutional neural networks approach for multimodal biometric identification system using the fusion of fingerprint, finger-vein and face images

El mehdi Cherrat[1], Rachid Alaoui[2,3] and Hassane Bouzahir[1]

[1] Laboratory of Systems Engineering and Information Technology, National School of Applied Sciences, Ibn Zohr University, Agadir, Morocco
[2] Laboratory of Computer Science and Telecommunications Research, Faculty of Sciences, Mohammed V University, Rabat, Morocco
[3] Multimedia, Signal and Communications Systems Team, National Institute of Posts and Telecommunication, Rabat, Morocco

## ABSTRACT

In recent years, the need for security of personal data is becoming progressively important. In this regard, the identification system based on fusion of multibiometric is most recommended for significantly improving and achieving the high performance accuracy. The main purpose of this paper is to propose a hybrid system of combining the effect of tree efficient models: Convolutional neural network (CNN), Softmax and Random forest (RF) classifier based on multi-biometric fingerprint, finger-vein and face identification system. In conventional fingerprint system, image pre-processed is applied to separate the foreground and background region based on $K$-means and DBSCAN algorithm. Furthermore, the features are extracted using CNNs and dropout approach, after that, the Softmax performs as a recognizer. In conventional fingervein system, the region of interest image contrast enhancement using exposure fusion framework is input into the CNNs model. Moreover, the RF classifier is proposed for classification. In conventional face system, the CNNs architecture and Softmax are required to generate face feature vectors and classify personal recognition. The score provided by these systems is combined for improving Human identification. The proposed algorithm is evaluated on publicly available SDUMLA-HMT real multimodal biometric database using a GPU based implementation. Experimental results on the datasets has shown significant capability for identification biometric system. The proposed work can offer an accurate and efficient matching compared with other system based on unimodal, bimodal, multimodal characteristics.

## INTRODUCTION

Biometric authentication system is basically a pattern-recognition system that identifies a human using a feature vector involved in a particular measurable morphological or behavioral characteristic the individual acquires. The biometrics modalities are often unique, measurable or automatically validated or permanent (*Cherrat et al., 2017*).

Corresponding author
El mehdi Cherrat,
elmehdi.cherrat@edu.uiz.ac.ma

Fingerprint have become an essential biometric trait due to its uniqueness and invariant to every individual. This biometric modality is more used and acceptable by the users due to acquiring device is comparatively small. Moreover, the recognition accuracy is relatively very high to others biometric recognition system based on the retina, ear shape, iris, etc. (*Borra, Reddy & Reddy, 2018*).

The finger vein biometric modality is usually used in biometric recognition because of many advantages compared other modality, (1) it is simple and easy to use: easily acquired using sensor capable of capturing or the NIR (Near-Infrared) light source; (2) it is high security: the vein structure is hidden inside the skin and the possibility of spoof the human recognition system is very complex; (3) the veins of each individual are unique and different (*Yang & Zhang, 2012*). The fingervein recognition is based on human veins characteristic for identification or verification of the individual (*Tome, Vanoni & Marcel, 2014*). As result, human and computer performance on fingervein recognition is a studies topic with both scientific research value and widely application prospects (*Kang et al., 2019*).

Face recognition is a biometric recognition technology based on human facial feature information for identification or verification. The algorithms using facial recognition are sensitive to variance in facial expressions and accessories, uncontrolled illumination, poses. In this regard, human and computer performance on facial identification is a research topic with both scientific research value and widely application prospects (*Mane & Shah, 2019*).

In order to overcome the limitation concerned with the system based on one modality biometric, the multimodal biometric system increase the robustness and performance against the imposters attack and environment variations. This system is classified as multi-instance, multi-sensor, multi-algorithm, multi-modal and hybrid systems (*Walia et al., 2019*).

The general structure of biometric recognition system consists of four main stages. First, the acquisition of biometric trait is process of getting a digitalized image of a person using specific capturing device. Second, the pre-processing is allowed to improve overall quality of the captured image. Third, the features data are extracted using different algorithms. Finally, the matching of the extracted characteristics is generally applied in order to perform the recognition of the individual.

The multi-biometric recognition system combines a variety of biometric sources. The main advantage of multimodal system against traditional single biometric is achieving the recognition process more secure and accurate (*Unar, Seng & Abbasi, 2014*). In this regard, researches of multimodal biometric using finger-vein and face images are prevalent and essential recently.

The advantage of combining the fingerprint, finger veins and face is the ability to establish an image acquisition system which can capture fingerprint and finger-vein images simultaneously (they find in almost at the same place) and its devices are less expensive and easier to deploy. Moreover, the face is one of the most natural methods to identify an individual, it does not restrict the movement of the person and its deployment cost is relatively low.

The proposed method deploys the multimodal biometric recognition system that is combined the fingerprint, finger-vein and face images using convolutional neural networks

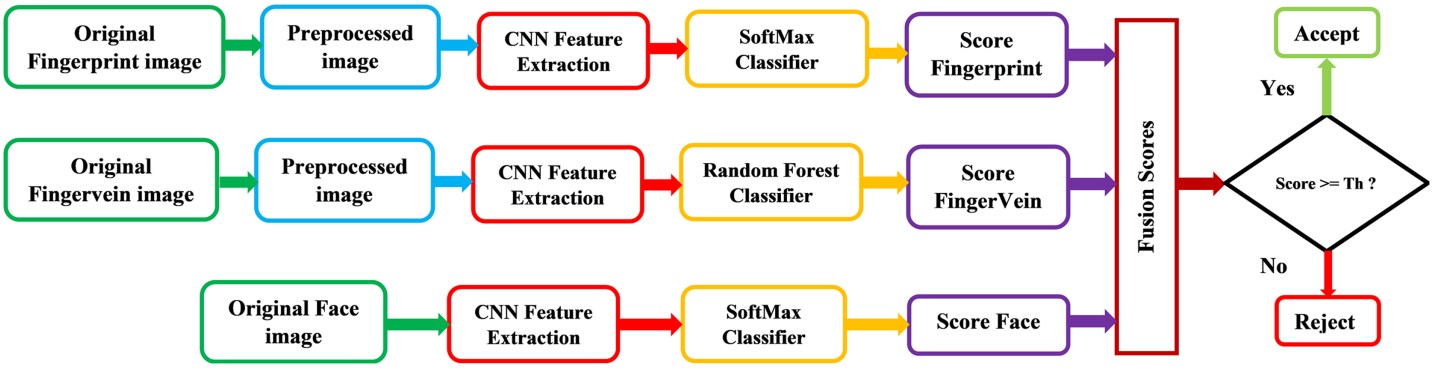

**Figure 1  General block diagram of the proposed recognition system.**

(CNNs) architectures and classifiers based on Softmax and Random forest (RF). Our scheme is efficient to various environmental changes and database types. Figure 1 describes general block diagram of the proposed recognition system.

The rest of the paper is separated as follows: Literature review defines a concise description of related research. Proposed system refers to the processes that are part of our proposed algorithm. While Experimental results and discussion elucidates the experimental results, Conclusion concludes the proposed work.

## LITERATURE REVIEW

Many studies has been conducted to investigate multimodal biometric system and its effects on the human recognition. *Ross & Jain (2003)* presented different levels of fusion and score level fusion on the multimodal biometric system (fingerprint, face, voice and hand geometry) using the sum rule. However, this method needs to conduct experiments on a larger database of users for recognition system. *Yang & Zhang (2012)* have been subjected a fusion of fingerprint and finger vein. These biometric characteristic is extracted using a unified Gabor filter method. The feature level fusion is generated based on supervised local preserving canonical correlation analysis framework. For individual identification, the nearest neighborhood classifier is applied. However, the performance is evaluated using a collected database which contains just 640 fingerprint images and 640 finger-vein images. *Son & Lee (2005)* have been subjected a fusion of face and iris using DWT and DLDA method. Each experiment is repeated at least 20 times for reducing the variation. Though, this algorithm is not compared with other state of the art methods. As well, it is not verified on a large number of data. *Ross & Govindarajan (2005)* presented multimodal biometric system that uses hand and face at feature level for biometric recognition purposes. Moreover, the experiments have been tested on intra-modal and inter-modal fusion with R, G, B channels. The drawbacks of this system, it does not accord incompatible feature sets (e.g., eigen-coefficients of face and minutiae points of fingerprints) to be combined and it is difficult to predict the best fusion strategy given a scenario. A novel fingerprint and finger vein identification system by concatenating the feature vectors are achieved by *Ma, Popoola & Sun (2015)*. In this study, the extracted

feature vectors of both fingerprint and finger vein images are concatenated in order to combine the classifiers recognition results at the decision level. Though, the accuracy of this technique does not satisfy the requirement of many real-world applications, where it suffers from significant performance translation and rotation invariant. *Huang et al. (2015)* introduced an adaptive bimodal sparse representation based on classification, that is, adaptive face and ear using bimodal recognition system based on sparse coding, where the qualities of weighted feature is selected. This system requires to pre-process each trait biometric before extracting the features. Furthermore, the recognition accuracy needs to increase.

*Yang et al. (2018)* presented a multi-biometric system cancelable using fingerprint and finger-vein, which combines the minutia points of fingerprint and finger-vein image feature based on a feature-level of three fusion techniques. However, the effect of noisy data on the performance of the system is not included. *Vishi & Mavroeidis (2018)* reported a fusion of fingerprint and finger-vein for identification using the combinations of score normalization (min-max, $z$-score, hyperbolic tangent) and fusion methods (minimum score, maximum score, simple sum, user weighting). The pre-processing stage is not used in this algorithm. Thus, the recognition accuracy can be decreased. *Jain, Hong & Kulkarni (1999)* introduced a multimodal biometric system using face, fingerprint and voice. Moreover, the different fusion techniques and normalization methods of fingerprint, hand geometry and face biometric sources are achieved by *Jain, Nandakumar & Ross (2005)*. The drawback of these methods, they need to be tested on a large dataset in real operating environment. *Soleymani et al. (2018)* are suggested a multimodal biometric system with face, iris and fingerprint using multiple streams of modality-specific CNNs. In this algorithm, some complexity also exists in multimodal recognition system which reduces its acceptability in many areas. Further, multimodal biometric system based on Iris, finger vein and fingerprint was investigated (*Walia et al., 2019*). In this method, individual classifier score estimation along with its performance optimization using evolutionary backtracking search optimization algorithm (BSA) is presented. In addition, the core design of the fusion model using proportional conflict redistribution rules (PCR-6) is proposed. The accuracy of 98.43% and an error rate of 1.57% have been achieved. However, the enhancement biometric quality and Experimentation with real multimodal dataset is not used in this system.

There exist only a few works about a multimodal biometric system that includes fingerprint, fingervein and face. *Rajesh & Selvarajan (2017)* are proposed an algorithm for biometric recognition using fingerprint and fingervein and face. They used score-level fusion to fuse these biometrics traits but they did not evaluate such a system against to others methods. As well, they have not provided information on the databases and the number of users in the study.

## PROPOSED SYSTEM

### Fingerprint recognition system

This section describes the detail about the proposed fingerprint recognition system using CNN-Softmax. In this work, our proposed method includes of the following

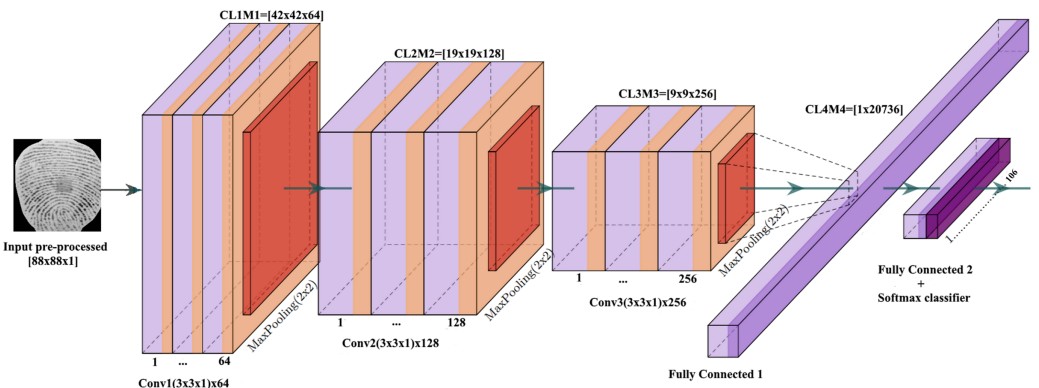

**Figure 2  The architecture of the proposed fingerprint-CNN model.**

three major stages: (1) pre-processing the fingerprint image; (2) feature extraction with CNN model; (3) using Softmax as a classifier. For pre-processing step, Soble and TopHat filtering method improved the quality of the image by limiting the contrast. After that, *K*-means and DBSCAN approaches are applied to classify the image into foreground and background region (*Cherrat, Alaoui & Bouzahir, 2019*). In addition, the Canny method (*Canny, 1987*) and the inner rectangle are adopted to extract the Region of interest (ROI) of fingerprint segmented. After this step, the features are extracted from the pre-processing fingerprint image using the CNN architecture.

The CNN is a convolutional neural network based on deep supervised learning model. In this regard, CNN can be viewed an automatic feature extractor and a trainable classifier (*Bhanu & Kumar, 2017*). As shown in Fig. 2, the configuration details of the proposed fingerprint-CNN architecture. The proposed model has five convolutional layers and three max-pooling layers which can be are computed using Eq. (1). In addition, three rectified linear unit (ReLU) are used to our system which can be defined as Eq. (2).

$$O_n = \sum_{i=1}^{N-1} x_i f_{n-i} \tag{1}$$

where *O* is the output map, *x* is input map, *f* is the filter and *N* is number of elements in *x*.

$$f(x) = \max(0, x) \tag{2}$$

where *x* is the input to a neuron.

The Softmax function can be used to the fully convolutional layer output, as shown in Eq. (3).

$$S(r, i) = -\log\left(\frac{e^{z_i}}{\sum_{k=1}^{N} e^{z_j}}\right) \tag{3}$$

when the vector of output neurons is set to $r$, the probability of the neurons appropriate to the $i^{th}$ class is provided by separation the value of the $i^{th}$ ($i = 1\dots j$) element by the sum of the values of all elements.

The structure is described as follows: (1) L1: the input layer data size of $88 \times 88$, which is the size of input pre-processing fingerprint images; (2) L1M1: first hidden layer, composed by 64 convolutional filters of size $3 \times 3 \times 1$, ReLU activation function and a max-pooling layer of size $2 \times 2$. This layer changes the input data into CL1M1 = ($42 \times 42 \times 64$) features; (3) L2M2: second hidden layer, composed by 128 convolutional filters of size $3 \times 3 \times 64$, ReLU activation function and a max-pooling layer of size $2 \times 2$. This layer changes the input data into CL2M2 = ($19 \times 19 \times 128$) features; (4) L3M3: third hidden layer, composed by 128 convolutional filter of size $3 \times 3 \times 128$, ReLU activation function and a max-pooling layer of size $2 \times 2$. In order to disconnect the connections between the first layer and the next layers the dropout probability (19) of 20% is adopted. This layer transforms the input data into CL3M3 = ($9 \times 9 \times 256$) features; (5) L4M4: forth hidden layer namely fully connected layer, represented the flattening process, which is converted all the resultant two-dimensional arrays into a single long continuous linear vector. The features size of input data is $1 \times 1 \times 20,736$; (6) L5M5: final hidden layer, this layer represented the feature descriptor of the finger vein for recognition to describe it with informative features. The Softmax function is used to predict labels of the input patterns.

## Fingervein recognition system

In this section, the proposed algorithm for fingervein recognition using CNN as a feature extractor is described. Our proposed method consist in three phases: (1) Canny method and the inner rectangle are used to obtain the ROI of finger vein image; (2) the exposure fusion framework (*Ying et al., 2017*) is applied to improve the contrast of the image by limiting the contrast amplification in the different region of the image. The result of finger vein image using Canny edge detector and contrast techniques such as contrast limited adaptive histogram equalization (*Reza, 2004*) and dynamic histogram equalization (*Abdullah-Al-Wadud et al., 2007*) is shown in Fig. 3. (3) feature extraction based on CNN and (4) RF is employed as a classifier for fingervein classification. The proposed model has five convolutional layers where three followed by max-pooling and three ReLU. The RF classifier is used to predict labels of the input patterns. Table 1 summarizes the characteristics of the proposed fingervein-CNN configuration.

## Random forest classifier

The RF algorithm proposed by *Breiman (2001)*, is an ensemble learning technique for regression and classification. At each split, RF consists of bagging (bootstrap aggregating) of $T$ decision trees with a randomized selection of features. Given a training data $X$, the RF algorithm is presented as follows: (i) at each $T$, generate a bootstrap sample with replacement from the original training data. (ii) Choose a random set of features using each bootstrap sample data, at each internal node. Furthermore, randomly select $Y$ predictors and pick the best split based on only the $Y$ predictors rather than all predictors. (iii) Aggregate the set of estimated decision trees in order to get a single one.
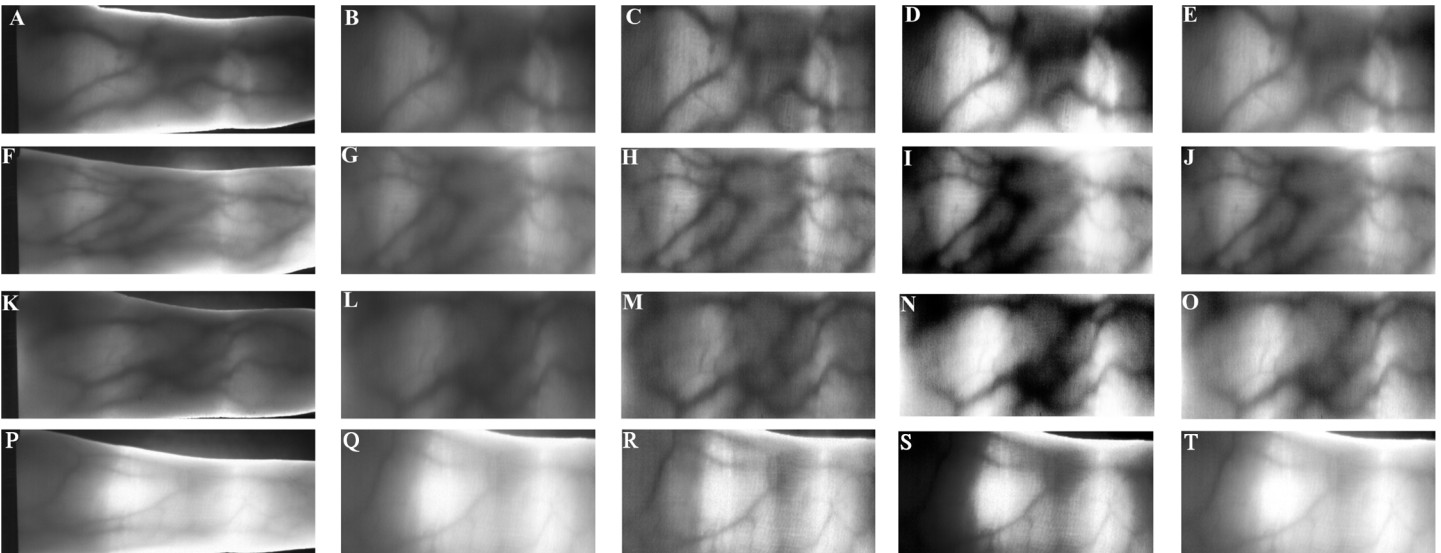

**Figure 3 Pre-processed finger-vein image using different enhanced algorithm from the Avera databases.** (A), (F), (K), (P) Original image. (B), (G), (L), (Q) Cropped image. (C), (H), (M), (R) CLAHE enhanced. (D), (I), (N), (S) DHE enhanced. (E), (J), (O), (T) Proposed enhanced using EFF.

**Table 1 Proposed fingervein-CNN configuration.**

| Type | Number of filter | Size of feature map | Filter size/stride |
| --- | --- | --- | --- |
| Convolution | 64 | $58 \times 150 \times 1$ | $3 \times 3/1$ |
| ReLU | – | $58 \times 150 \times 1$ | – |
| Max-pooling | 1 | $29 \times 75 \times 64$ | $3 \times 3/2$ |
| Convolution | 128 | $27 \times 73 \times 128$ | $3 \times 3/1$ |
| ReLU | – | $27 \times 73 \times 128$ | – |
| Max-pooling | 1 | $13 \times 36 \times 128$ | $3 \times 3/2$ |
| Convolution | 256 | $11 \times 34 \times 256$ | $3 \times 3/1$ |
| ReLU | – | $11 \times 34 \times 256$ | – |
| Max-pooling | 1 | $5 \times 17 \times 256$ | $3 \times 3/2$ |
| Fully-connected | 1 | $1 \times 21760$ | – |
| Fully-connected | 1 | $1 \times 106$ | – |

## Face recognition system

In this section, the proposed algorithm for face recognition using CNN as a feature extractor is described. Our proposed method consist in two phases: feature extraction based on CNN and employing Softmax as a classifier for face classification. Table 2 shows the configuration details of the proposed CNN architecture using face image. The proposed model has five convolutional layers where three followed by max-pooling and three ReLU. In order to disconnect the connections between the first layer and the next layers the dropout probability (*Srivastava et al., 2014*) of 20% is adopted. In addition, the dropout probability of 10% between the second layer and the next layers.

**Table 2 Proposed face-CNN configuration.**

| Type | Number of filter | Size of feature map | Filter size/stride |
|---|---|---|---|
| Convolution | 32 | $88 \times 88 \times 1$ | $3 \times 3/1$ |
| ReLU | – | $88 \times 88 \times 1$ | – |
| Max-pooling | 1 | $44 \times 44 \times 32$ | $3 \times 3/2$ |
| Convolution | 64 | $42 \times 42 \times 64$ | $3 \times 3/1$ |
| ReLU | – | $42 \times 42 \times 64$ | – |
| Max-pooling | 1 | $21 \times 21 \times 64$ | $3 \times 3/2$ |
| Convolution | 128 | $19 \times 19 \times 128$ | $3 \times 3/1$ |
| ReLU | – | $19 \times 19 \times 128$ | – |
| Max-pooling | 1 | $9 \times 9 \times 128$ | $3 \times 3/2$ |
| Fully-connected | 1 | $1 \times 10388$ | – |
| Fully-connected | 1 | $1 \times 106$ | – |

## Feature extraction fusion

In this section, we introduce our proposed method score level fusion technique based on the matching score level fusion. This matching score indicates better proximity of characteristic vector with the template.

The fused score level is based on the weighted sum and weighted product as shown in Eqs. (4) and (5). If the fused score value providing of the query fingerprint, finger vein and face is greater than or equal to the decision threshold value. Then, the person is accepted, otherwise is rejected (*Singh, Singh & Ross, 2019*).

$$\text{Score}_{\text{ws}} = w_1 S_{\text{FP}} + w_2 S_{\text{FV}} + w_3 S_{\text{FA}} \tag{4}$$

$$\text{Score}_{\text{wp}} = S_{\text{FP}}{}^{w_1} \times S_{\text{FV}}{}^{w_2} \times S_{\text{FA}}{}^{w_3} \tag{5}$$

where $S_{\text{FP}}$, $S_{\text{FV}}$, $S_{\text{FA}}$ indicate the scores of biometric matcher, $w_1$, $w_2$, $w_3$ are the weight value over a range of (0, 1) and according to sum of $w_1$, $w_2$, $w_3$ is always 1.

## Data augmentation

Data augmentation is one of the methods for reducing the effects of overfitting problems in CNN architecture. This technique is employed to increase the amount of training data based on image translation, rotation and cropping process. Many previous works have been successfully used data-augmentation method. We implemented the data augmentation as expand to the work in *Krizhevsky, Sutskever & Hinton (2012)* such as the rotation and the translation (left, right, up and down) (*Park et al., 2016*). For SDUMLA-HMT database augmentation, we were augmented that is two times larger than the original database.

## EXPERIMENTAL RESULTS AND DISCUSSION

The experimental operation platform in this study is described as follows: the host configuration: Intel Core i7 − 4770 processor, 8Go RAM and NVIDIA GeForce GTX 980 4GO GPU, runtime environment: Ubuntu 14.04 LTS (64 bit). In order to better verify our algorithm, the following classification methods are adopted in the experiment: support

**Table 3 Dataset structure of fingerprint, fingervein and face databases.**

|  | SDUMLA-HMT database |
|---|---|
| Class number | 106 |
| Image number | 41,340 |
| Training | 33,072 |
| Validation | 4,134 |
| Test | 4,134 |

**Table 4 The training set result of proposed fingerprint recognition using CNN.**

| Images | Train set without dropout | | Training set with dropout | |
|---|---|---|---|---|
|  | Accuracy (%) | Loss (%) | Accuracy (%) | Loss (%) |
| Original$_{FP}$ | 98.96 | 3.65 | 99.31 | 2.35 |
| Enhanced$_{FP}$ | 99.49 | 1.93 | 99.56 | 1.23 |
| Proposed Enhanced$_{FP}$ | 99.13 | 2.16 | 99.63 | 1.17 |

vector machine (SVM) (*Cortes & Vapnik, 1995*), RF (*Breiman, 2001*), logistic regression (LR) (*Hosmer, Lemeshow & Sturdivant, 2013*), fingervein biometric system (*Itqan et al., 2016*) and Multimodal biometric system using fingerprint, fingervein and face (*Rajesh & Selvarajan, 2017*). These algorithms were compared to each other. In order to validate the proposed algorithm, the results have been tested on the public on SDUMLA-HMT (*Yin, Liu & Sun, 2011*) database which includes real multimodal data of fingerprint, fingervein and face images. The total number of training images was 41,340 and we divided them into training, validation and test sets. The divided data set used in the experiment is shown in Table 3.

The performance measure is accuracy rate as defined by Eq. (6).

$$\text{Accuracy} = \frac{TP + TN}{TP + TN + FP + FN} \times 100 \tag{6}$$

where True Positive Rate (TP) is the probability of authorized users that are recognized correctly over the total number tested, True negative rate (TN) is the probability of authorized users that are not recognized over the total number tested. False positive rate (FP) describes the percentage of unauthorized users that are recognized to the total number tested. False negative rate (FN) describes the percentage of unauthorized users that are not recognized falsely to the total number tested.

As can be seen from Table 4, the proposed fingerprint recognition using CNN with dropout method leads to a significant performance improvement on real database multimodal.

In particular, the highest accuracy gain was obtained by dropout method on four datasets for training and test set. Moreover, the least loss is achieved based on database using the dropout method. For training set, it can be noted from the Table 4 that the accuracy of 99.13% is augmented to 99.63% and the lost rate of 2.16% is reduced to 1.17%

**Table 5 The test set result of proposed fingerprint recognition using CNN.**

| Images | Test set without dropout | | Test set with dropout | |
|---|---|---|---|---|
| | Accuracy (%) | Loss (%) | Accuracy (%) | Loss (%) |
| Original$_{FP}$ | 97.06 | 9.12 | 97.66 | 5.71 |
| Enhanced$_{FP}$ | 98.29 | 5.68 | 99.16 | 3.14 |
| Proposed enhanced$_{FP}$ | 99.33 | 2.16 | 99.48 | 2.03 |

**Table 6 The training set result of proposed fingervein recognition using CNN.**

| Images | Accuracy (%) | Loss (%) |
|---|---|---|
| Original$_{FV}$ | 96.58 | 19.12 |
| Cropped$_{FV}$ | 97.09 | 15.22 |
| CLAHE$_{FV}$ | 98.45 | 6.36 |
| DHE$_{FV}$ | 97.65 | 11.12 |
| Proposed enhanced$_{FV}$ | 99.09 | 2.69 |

**Table 7 The test set result of proposed fingervein recognition using CNN.**

| Images | Accuracy (%) | Loss (%) |
|---|---|---|
| Original$_{FV}$ | 96.98 | 12.08 |
| Cropped$_{FV}$ | 97.89 | 9.32 |
| CLAHE$_{FV}$ | 98.97 | 4.23 |
| DHE$_{FV}$ | 98.25 | 5.52 |
| Proposed enhanced$_{FV}$ | 99.27 | 2.05 |

**Table 8 The training set result of proposed face recognition using CNN.**

| Images | Train set without dropout | | Training set with dropout | |
|---|---|---|---|---|
| | Accuracy (%) | Loss (%) | Accuracy (%) | Loss (%) |
| Original$_{Fa}$ | 99.25 | 1.96 | 99.55 | 1.77 |

in the proposed method due to add the drop function in our system. For test set, based on the results yielded in Table 5, the accuracy of 99.33% is augmented to 99.48% and the lost rate of 2.16% to 2.03% in the proposed fingerprint identification method.

The comparison of pre-processed algorithms with the proposed fingervein system are shown in Tables 6 and 7. The proposed work gives the highest average accuracy for training set with 99.09% and least loss with 2.69%. For test set, also the highest average accuracy of 99.27% and least loss with 2.05 is obtained using our algorithm.

The results of Tables 8 and 9 show that dropout method plays an important role in increasing the accuracy of the proposed face recognition system. For training set, the accuracy of 99.25% is augmented to 99.55% and the lost rate of 1.96% is lowered to 1.77%. For test set, the accuracy of 99.05% is augmented to 99.13% and the lost rate of 2.10% is lowered to 2.17%.

**Table 9  The test set result of proposed face recognition using CNN.**

| Images | Test set without Dropout | | Test set with dropout | |
|---|---|---|---|---|
| | Accuracy (%) | Loss (%) | Accuracy (%) | Loss (%) |
| Original$_{Fa}$ | 99.05 | 2.10 | 99.13 | 2.27 |

**Table 10  The result of proposed system recognition unimodal biometric using CNN with different classifiers.**

| Classifiers | Fingerprint | Finger vein | Face |
|---|---|---|---|
| CNN & SoftMax | 99.48% | 99.27% | 99.13% |
| CNN & SVM | 97.65% | 99.33% | 97.88% |
| CNN & LR | 85.61% | 84.14% | 92.43% |
| CNN & RF | 97.33% | 99.53% | 91.95% |

**Table 11  The result of proposed recognition systems using CNN with rules fusion.**

| Algorithms | Rules fusion | |
|---|---|---|
| | Weighted sum | Weighted product |
| Fingerprint$_{CNN}$ & Fingervein$_{CNN}$ | 99.59 | 99.58 |
| Fingerprint$_{CNN}$ & Face$_{CNN}$ | 99.30 | 99.28 |
| Fingervein$_{CNN}$ & Face$_{CNN}$ | 99.20 | 99.17 |
| Fingerprint$_{CNN}$ & Fingervein$_{CNN}$ & Face$_{CNN}$ | 99.73 | 99.70 |

**Table 12  Computational time (ms) for fusion method of database.**

| Algorithms | Time (ms) |
|---|---|
| Fingerprint$_{ANN}$ & Fingervein$_{ANN}$ & Face$_{ANN}$ (*Rajesh & Selvarajan, 2017*) | 130 |
| Score level fusion model (*Walia et al., 2019*) | 580 |
| Proposed fingerprint$_{CNN}$ & Fingervein$_{CNN}$ & Face$_{CNN}$ | 69 |

From Table 10, we compared the results using the SVM, LR, RF with CNN using the proposed fingerprint system to show the highest accuracy of 99.48% is tacked using the Softmax classifier. As shown in this Table, the RF classifier gives the highest accuracy of 99.53% using the proposed fingervein system using CNN architecture. Also based on the results yielded in Table 10 it can be argued that the Softmax classifier gives the highest accuracy of 99.13% based on the proposed face system using the CNN model.

From Table 11, it is clear that the highest recognition rate is obtained when weighted sum is used for fusion rule. Table 12 presents computational time for fusion method of database. Table 13 shows the comparison between the proposed system unimodal, bimodal and multimodal biometric that using CNN on database. The proposed fingerprint, fingervein and face as a bimodal system can be used for recognition with acceptable identification results comparing with other unimodal systems. The proposed multimodal biometric system is increased the recognition accuracy than the unimodal and

**Table 13 The accuracy rate for proposed systems and different recognition biometric system results.**

| Algorithms | Accuracy (%) |
| --- | --- |
| Enhanced fingerprint$_{CNN}$ using *Cherrat, Alaoui & Bouzahir (2019)* | 99.48 |
| Enhanced fingervein$_{CNN}$ using *Ying et al. (2017)* | 99.53 |
| Fingervein$_{CNN}$ (*Itqan et al., 2016*) | 96.65 |
| Proposed face$_{CNN}$ | 99.13 |
| Proposed fingerprint$_{CNN}$ & Fingervein$_{CNN}$ | 99.51 |
| Proposed fingerprint$_{CNN}$ & Face$_{CNN}$ | 99.31 |
| Proposed fingervein$_{CNN}$ & Face$_{CNN}$ | 99.33 |
| Fingerprint$_{ANN}$ & Fingervein$_{ANN}$ & Face$_{ANN}$ (*Rajesh & Selvarajan, 2017*) | 99.23 |
| Score level fusion model (*Walia et al., 2019*) | 99.61 |
| Proposed fingerprint$_{CNN}$ & Fingervein$_{CNN}$ & Face$_{CNN}$ | 99.49 |

bimodal identification system, where the accuracy rate is 99.49%. Although existing biometric method (*Walia et al., 2019*) is able to obtain 99.61% of recognition rate, it is still slower than our proposed fusion method in terms of computational time of 69 ms.

Finally, we can conclude from these results that the proposed multimodal system is superior to other methods because:

1. The proposed enhanced fingerprint and finger vein patterns are significantly clearly distinguishable and more prominent in their others enhanced versions. Therefore, the proposed methods are typically able to guarantee a high identification rate.
2. The recognition accuracy based on dropout method is better than using only the dataset method.
3. CNN approach can usually provide better performances than using combinations between different processes such as windowing, extracting features, etc. Thus, the recognition biometric system based on CNN technique can surpassed other classical and complicated techniques.
4. The proposed multimodal algorithm have higher accuracy to identify the person and ensure that its information or data is safer compared to system based on single or bimodal biometrics.

## CONCLUSION

A system for human recognition using CNN models and a multimodal biometric identification system based on the fusion of fingerprint, fingervein and face images has been introduced in this work. The experimental results on real multimodal database have shown that the overall performance of the proposed multimodal system is better than unimodal and bimodal biometric systems based on CNN and different classifiers regarding identification. From the results obtained, we can also conclude that the effect of the pre-processed algorithm improved the accuracy rate of the proposed system. Dropout technique plays an important role for increasing the recognition accuracy, which reduced the loss rate of the system. For future study, extending the proposed algorithm to

other applications is a task worth investigating, where it will be tested with a more challenging dataset that contains a large number of subjects.

### Funding
The authors received no funding for this work.

### Competing Interests
The authors declare that they have no competing interests.

### Author Contributions
- El mehdi Cherrat analyzed the data, performed the computation work, conceived and designed the experiments, performed the experiments, prepared figures and/or tables, authored or reviewed drafts of the paper, and approved the final draft.
- Rachid Alaoui analyzed the data, performed the computation work, conceived and designed the experiments, performed the experiments, prepared figures and/or tables, authored or reviewed drafts of the paper, and approved the final draft.
- Hassane Bouzahir analyzed the data, performed the computation work, conceived and designed the experiments, performed the experiments, prepared figures and/or tables, authored or reviewed drafts of the paper, and approved the final draft.

### Data Availability
SDUMLA-HMT real multimodal database is available at: http://mla.sdu.edu.cn/info/1006/1195.htm.

Code and data are available in the Supplemental Files.

### Supplemental Information
Supplemental information for this article can be found online at http://dx.doi.org/10.7717/peerj-cs.248#supplemental-information.

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
