# Peer review of "Convolutional neural networks approach for multimodal biometric identification system using the fusion of fingerprint, finger-vein and face images"

_PeerJ Computer Science, doi:10.7717/peerj-cs.248_

## Round 0.1 · original submission · Major Revisions

The literature review is inadequate. Reviewer 2 identifies some references that should be cited; Reviewer 1 notes that there is a lack of analysis of the literature. The literature should be carefully analyzed and the contribution of the paper should be defined in relation to the existing literature. In particular, there is now a substantial literature on multimodal biometric identification. The authors need to explain the advantages of combining these three features specifically.

Please also consider carefully the suggestions made by Reviewer 2 for improving the writing and clarity.

Reviewer 1 ·

Basic reporting

Literature review is just a pile of information, lacking of analysis and induction.

Experimental design

Authors proposed an approach for multimodal biometric identification system using the fusion of fingerprint, finger-vein and face images to overcome the short comings of single biometric and achieve the high-performance accuracy, and explain the validity of the multimodal biometric system by conducting some experiments.

Validity of the findings

Only combining Fingerprint, Finger-Vein and Face Images lack of some innovation, since a lot of multimodal biometric identification system have been proposed (just as mentioned in "Literature review"), authors should state clearly the advantage of combining the three traits, not just pick up 3 traits randomly.

Additional comments

Authors proposed an approach for multimodal biometric identification system using the fusion of fingerprint, finger-vein and face images to overcome the short comings of single biometric and achieve the high-performance accuracy, and explain the validity of the multimodal biometric system by conducting some experiments.

But in fact, only combining Fingerprint, Finger-Vein and Face Images lack of some innovation, since a lot of multimodal biometric identification system have been proposed (just as mentioned in "Literature review"), authors should state clearly the advantage of combining the three traits, not just pick up 3 traits randomly.

Literature review is just a pile of information, lacking of analysis and induction.

in line 192-193, the author wrote " where k indicates the biometric matcher, w1 ,w2 ,……. wk are the weight value over a range of [0,1] and according to the condition and the sum of w1,w2,……,wk is always 1."
since K=3, authors should rewrite the sentence (use 3 replace K, and adjust the presentation of the sentence) to make it more accurate and clearer

in line 170-171, the author wrote "decision trees in order to get a single decision tree. Figure 4 presents an example of separation of classes with the Random Forest. The RF algorithm is given in Fig. 5."
in fact, it is superfluous to present an example of separation of classes with the Random Forest, since Random Forest is well known method. in addition, we cannot find the Fig. 5 the author mentioned.

Reviewer 2 ·

Basic reporting

The paper address issues of unimodal biometric system by fusion of multiple modalities. Approach seems to be technically good. It is difficulty to forge multiple modalities and hence can be more secure. The literature review needs to be augmented with some of cancellable approach which also impart security of template. these papers needs to be added in literature review

1. "Secure multimodal biometric system based on diffused graphs and optimal score fusion", Volume 8, Issue 4, July 2019, p. 231 – 242, IET biometric.
2. ‘A review of biometric technology along with trends and prospects’, Pattern Recognit., 2014, 47, (8), pp. 2673–2688.
3. ‘An adaptive bimodal recognition framework using sparse coding for face and ear’, Pattern Recognit. Lett., 2015, 53, pp. 69–76.

Results are sufficient to prove the efficiency of method. I suggest to include the recognition index in results section.

There are number of grammatical errors, Sentence format needs to be improved. Try to write short and concise sentence for better understanding of readers.

Experimental design

Comparison with other state of the art methods needs to included in this section.

Validity of the findings

Seems okay

---

## Round 0.2 · Major Revisions

The reviewer indicates that the revision partially addresses the reviewer's concerns. Several other important problems remain with the paper:

1. Please discuss the reference suggested by the reviewer in the literature review.
2. The reviewer asks for results on real data and comparison of the suggested score fusion method.
3. The results tables (4-14) give a single number for each method, measure, and data set. There is no way for the reader to judge whether the differences between the methods are real or statistical artifacts. There should be some statistical analysis to help us judge the meaning of these differences. This is the standard now in machine learning research.

Reviewer 2 ·

Basic reporting

The authors has partially addressed my previous comments. Although approach seems to be good. But authors needs to addressed the comments before final acceptance of manuscripts. I strongly feel the comparison with state of the art score fusion should be included in Table 14 as in current submission compared multimodal papers are not very recent. I suggest to included score fusion comparison in table 14

Robust multimodal biometric system based on optimal score level fusion model, Expert Systems with Applications, Volume 116, February 2019, Pages 364-376

Experimental design

Experimentation with real multimodal dataset can be included rather that virtual multimodal database.
Some ablation analysis about the proposed method should further strengthen the claims.

Validity of the findings

Seems okay but experimentation with real multimodal database needs to be included.

Additional comments

Experiments with real multimodal database.
Comparison with state of the art score fusion method.

---

## Round 0.3 · accepted · Accept

We have received a response from one of the reviewers of the previous versions of your paper, who has indicated that your paper has satisfied all the concerns identified by the reviewer. Therefore, the paper is accepted for publication. I thank you for your interest in PeerJ and congratulate you on your successful publication.

Reviewer 2 ·

Basic reporting

The author has address most of my earlier comments. The approach can be accepted for publicatin

Experimental design

Sufficient

Validity of the findings

Can be further augmented by comparing with state of the art methods